# How Do Elite Female Athletes Cope with Symptoms of Their Premenstrual Period? A Study on Rugby Union and Football Players’ Perceived Physical Ability and Well-Being

**DOI:** 10.3390/ijerph191811168

**Published:** 2022-09-06

**Authors:** Roberto Modena, Elisa Bisagno, Federico Schena, Simone Carazzato, Francesca Vitali

**Affiliations:** 1CeRiSM Research Center Sport Mountain and Health, University of Verona, 37129 Verona, Italy; 2Faculty of Health Sciences and Social Care, Molde University College, 6410 Molde, Norway; 3Department of Law, University of Modena and Reggio Emilia, 42121 Modena, Italy; 4Department of Neurosciences, Biomedicine and Movement Sciences, Section of Movement Sciences, University of Verona, 37131 Verona, Italy

**Keywords:** coping strategies, premenstrual symptoms, athletes’ wellness, team sports

## Abstract

Women’s participation in sports has recently grown worldwide, including in sports typically associated with men (e.g., rugby and football). Similarly, literature on female athletes has increased, but how they cope with premenstrual (PM) physical and affective symptoms remains a poorly studied topic. Our study aimed to explain which coping strategies elite female rugby and football players use during their PM period to maintain perceived physical ability (PPA) and well-being. A mediation model analysis considering coping strategies (i.e., avoiding harm, awareness and acceptance, adjusting energy, self-care, and communicating) as independent variables, PPA and well-being as dependent variables, and PM physical and affective symptoms and PM cognitive resources as mediators was run on the data collected via an anonymous online survey. A dysfunctional impact of avoiding harm (indirect) and adjusting energy (both direct and indirect) and a functional indirect influence of awareness and acceptance, self-care, and communicating as coping strategies were found on PPA and well-being during the PM period. As predicted, PM physical and affective symptoms as mediators reduced PPA and well-being, while PM cognitive resources enhanced them. These results may inform practitioners on how to support elite female athletes’ PPA and well-being by knowing and reinforcing the most functional PM coping strategies for them.

## 1. Introduction

Women’s participation in sports has grown worldwide over the last decades [1]. This increasing trend is also true in several sports that are typically associated with men such as rugby and football, two of the most popular team sports worldwide. Indeed, World Rugby considered women’s participation as the fastest growing area for rugby promotion [2]. In addition, from 2014 to 2018, the number of female rugby players has increased by 51%, with the number of those registered rising by 145%, according to World Rugby. Today, more than a quarter (27%) of the 9.6 million athletes who play rugby in World Rugby member unions are female athletes [3]. Likewise, Fédération Internationale de Football Association (FIFA) considered women’s football as the single largest growth opportunity in football nowadays, and enhancing female participation in football was set as a top priority [4,5,6].

This new attention to female sport is slowly being reflected in sports research. Bonato et al. [7] suggested sport sciences to adopt the gender perspective as the mainstream one (gender mainstreaming can be described as the integration of the gender equality perspective into every stage of development and implementation of research, policy, or program) when researching and presenting evidence-based knowledge to practitioners. Indeed, given the numerous differences (e.g., physiological, biochemical, biomechanical, anatomical, psychological) between female and male athletes [8,9,10,11], only a small part of the findings of studies involving male athletes can be applied to female athletes [12].

One difference between female and male athletes is represented by the menstrual cycle (MC), the female biological rhythm characterized by fluctuations in endogenous steroid hormones, such as oestrogen and progesterone. These hormonal fluctuations are used to discriminate between three relevant phases of the MC: the early follicular phase (low oestrogen and progesterone), the late follicular phase (high oestrogen and low progesterone), and the mid-luteal phase (high oestrogen and progesterone). Although some studies [13,14] have shown that female athletes’ performance may be influenced by these hormonal fluctuations, results are often inconsistent, and a consensus is not established [13].

The 5–7 days before the menses represent the premenstrual (PM) period. Evidence has shown that the course and magnitude of the PM syndrome are associated with different physical (e.g., abdominal cramps, headaches, reduced energy levels) and psychological symptoms (e.g., worry, distraction, negative mood states) [14]. Psychological symptoms can be related to the affective (e.g., negative mood states) and/or cognitive (e.g., distraction) spheres and, together with physical symptoms, may negatively impact both the well-being and performance of female athletes during the PM period [15,16].

Strategies and their efficacy in dealing with dysfunctional physical and psychological symptoms related to the PM period have been studied in the general female population. According to the contextual cognitive model, coping is defined as a strategy and a way of thinking, feeling, and behaving that can be used by a person to cope with a challenging situation that is appraised as significant and difficult [17]. Research on PM coping strategies has studied the relationships between them and the severity of PM symptoms [18,19], as well as their association with PM-related depression, anxiety, and dysphoria [20,21]. Evidence reveals that coping strategies vary across the MC, with emotion-focused coping increasing, and task avoidance, as well as social diversion-oriented coping, decreasing in the PM period [22]. Women who show higher rates of PM-related depression engage in avoiding coping independently from the MC phase [19]. Besides, support from family and friends is associated with lower PM symptoms and more effective coping [23,24].

A qualitative study showed that women with moderate and severe PM symptoms adopt a wide range of coping strategies to reduce or avoid distress, including anticipation and planning, avoiding stress, self-care, solitude, but also expressing anger or irritation, seeking social support, taking supplements or drugs [22]. In addition, women who were more aware and accepted the PM symptoms were more likely to adopt awareness and acceptance as coping strategies.

Nevertheless, up to now, very few studies have explored the specific association between PM symptoms and coping strategies, and, in most of these studies, coping has been measured using generic coping scales rather than a specific PM coping measure [14]. Moreover, most studies examined how the general female population copes with life stressors rather than PM symptoms themselves. On top of this, the individual perceptions and coping strategies of elite female athletes regarding specific PM-related symptoms in the sports context still remain poorly known.

An exception is a qualitative study based on a semi-structured interview conducted by Findlay et al. [25] which involved 15 international female rugby players; almost all the participants reported having negative physical or physiological symptoms, mainly a few days before and at the onset of menses. Moreover, these elite female rugby players reported higher negative perceptions about physical symptoms during training compared to matches. Interestingly, these participants confirmed their discomfort to talk about this topic with men either belonging to the staff or not. In another qualitative study, Brown et al. [26] confirmed the aforementioned findings on 17 elite female athletes from different sports. Furthermore, almost all the participants declared managing the negative feelings, trying to accept them or adapt to them.

The overall findings showed the perception of a negative impact of the PM-specific symptoms on elite female athletes’ well-being and readiness to train and compete. Nevertheless, the knowledge and use of PM coping strategies by elite female athletes seem to be very poor, though elite female athletes understand the importance of doing something to improve their well-being and performance during this specific period [25,26].

Most of the few recent studies on PM symptoms in the sports context involved a small number of participants and used a descriptive qualitative approach [26]. Moreover, one of the topics that have been relatively poorly studied is how elite female athletes cope with specific PM physical and psychological symptoms (for an exception see [27]) and how these coping strategies impact their perceived performance and well-being.

Our study aimed to explain which PM coping strategies elite female rugby and football players use during their specific PM period to maintain perceived physical ability (PPA) and well-being by using a quantitative approach. PPA is a concept close to Bandura’s self-efficacy [23,24,25,26,27,28,29,30], which represents a cognitive mechanism, and a motivational dimension which mediates perception of personal capacities, and the individual conviction to achieve specific levels of performance. Self-efficacy related to sports activities has been widely verified [31,32,33,34,35,36] and constitutes a frame of reference to explain relationships between cognitive processes and physical performance.

Well-being represents a psychological dimension based on the principle that “health is not merely the absence of disease or infirmity, but a biopsychosocial state of complete physical, mental, and social well-being” [37]. The importance of optimal well-being and mental health in elite athletes has received increasing attention [38]. A particular challenge for an elite athlete is, therefore, to protect and stimulate her/his well-being in the highly demanding and performance-oriented elite sports context [39].

Moving from Martin et al. [40], who showed that elite female athletes abstain from exercise and training (4.2% of the cases) less often than the general female population from daily activities (15–29% of the cases), we designed our hypotheses on the functionality of coping strategies for athletes’ PPA and well-being. In this sense, the pressure to train, compete, and perform might be the cause of this difference: if for the general female population avoiding harm and adjusting energy as coping strategies might be beneficial to reduce specific PM distress, for elite female athletes, avoidance of distress is not always possible, as well as adjusting energy by regulating or reducing exercise. Therefore, we developed three main hypotheses for our study as follows:

**Hypothesis** **1** **(H1).***The avoiding harm and adjusting energy PM coping strategies should be negatively associated with the participants’ PPA and well-being. This is not only because these PM coping strategies might indicate neglect of the problem or complete change of habits that could be dysfunctional for elite female athlete’s needs [26], but also because for an elite female athlete, it is not always possible to avoid situations and people causing PM symptoms and refrain from exercise*.

**Hypothesis** **2** **(H2).***Awareness and acceptance, self-care, and communicating should be PM coping strategies positively associated with the participants’ PPA and well-being because these PM coping strategies indicate either positive disengagement or agency on the problem*.

**Hypothesis** **3** **(H3).**
*PM physical and affective symptoms and available PM cognitive resources should act as the mediators of the relationship between all the PM coping strategies and the outcome variables (i.e., PPA and well-being). This hypothesis moves from the idea that PM coping strategies reduce negative PM physical and affective symptoms and empower PM cognitive resources, and this, in turn, might enhance elite female athletes’ PPA and well-being.*


## 2. Materials and Methods

### 2.1. Study Design

A mediation model analysis considering coping strategies (i.e., avoiding harm, awareness and acceptance, adjusting energy, self-care, and communicating) as independent variables, PPA and well-being as dependent variables, and PM physical and affective symptoms and PM cognitive resources as mediators was run.

### 2.2. Eligibility Criteria and Sample Characteristics

The eligibility criteria to participate in this study were as follows: (1) female rugby union or football player; (2) aged 16 years or more; (3) elite female athlete; and (4) active player at the moment of the survey that is not retired or temporarily out of sports (e.g., due to a sports injury, motherhood). The age range of the participants was selected to guarantee an adequate distance from the menarche and for the participants to be self-aware about their MC, and specifically about the physical and affective symptoms and cognitive resources related to their PM period. We aimed to involve the topflight in each sport (i.e., rugby and football) to guarantee sufficient expertise of the participants and comparable standards of training and performance.

### 2.3. Procedure

Data were collected through an anonymous online quantitative survey (Google Forms, Google, Mountain View, CA, USA). The time required to complete the survey took approximately 20 min. Female rugby players were recruited by the Italian Rugby Federation, which in turn put the last author in contact with all the clubs in the Italian women’s rugby union, female football players—through direct contacting by the first author of the Italian Serie A and B female football clubs. All the participants under 18 years of age provided a free written consent of both parents to participate, while elite female athletes aged 18 years or more signed a free written consent, in both cases after receiving a full description of the protocol of the study and their rights to anonymity. All the procedures were conducted in accordance with the Declaration of Helsinki, and the Institutional Ethics Committee of the University of Verona provided ethical approval (protocol code 38.R1/2021; date of approval: 14 April 2022).

### 2.4. Measures

The online questionnaire was administered in Italian to the participants during the months of May and June 2022 through a link (https://forms.gle/PLEtoXdVue8nQ7An7). The participants completed it on their mobile phones at their respective training centres under the supervision of their coaches, previously informed and trained on the research. The survey consisted of two parts: the first one collected sociodemographic data (i.e., age, sports experience, deliberate practice) and information related to MC (i.e., contraceptives used, the average length of the MC, time from menarche, and the number of children), in addition to gender preferences regarding discussion of PM issues with the coach/staff members or teammates. The second part of the survey included several scales, preceded by the headline referring to the PM period, for the participants to frame their answers. The scales included in this survey are detailed below.

#### 2.4.1. PM Coping Strategies

PM coping strategies were assessed using a specific PM coping measure, namely the Premenstrual Coping Measure (PMCM) [14], which was adapted to the sports context.

The scale consists of thirty-two items and five dimensions, which represent different PM coping strategies. Avoiding harm regards avoiding situations, people, interactions, conversations, and thoughts which can cause distress when the athlete feels more premenstrually sensitive (e.g., “I avoid situations that have the potential to provoke me”). Awareness and acceptance refer to the awareness of the need to adopt coping strategies to deal with specific PM-related discomfort and accept PM changes as a natural part of a woman’s experience (e.g., “I accept my changeable moods”). Adjusting energy is based on behaviors aimed at regulating physical and psychological states, such as decreasing social activities, but also changing the habits regarding eating behavior and regulating exercise (e.g., “I adjust the training’s intensity or volume”). Self-care is the engagement in relaxing activities (e.g., “I take time to focus on my own needs”). Lastly, communicating (e.g., “I feel confident to tell people how I feel”) is based on seeking support and telling others about personal feelings and needs in this specific phase of the MC. For each PM coping strategy presented, the participants were asked to grade them on a Likert scale ranging from 1 (strongly disagree) to 5 (strongly agree). We computed five scores by averaging the items related to each dimension. In our study, the internal consistency (Cronbach’s alpha) ranged from 0.64 to 0.88.

#### 2.4.2. PM Physical and Affective Symptoms

PM physical symptoms were assessed with the seven-items subscale of the Daily Symptoms Rating Scale (DSRS) [41], a list of bodily symptoms that women can experience during their PM period (e.g., “tender or painful breasts”). Each item was scored on a Likert scale ranging from 1 (not at all) to 5 (very much). A single score was computed by averaging the seven items (in our study, Cronbach’s alpha was 0.80), with higher scores indicating worse PM physical symptoms.

Similarly, PM affective symptoms were assessed using the 10-items subscale of the Daily Symptoms Rating Scale (DSRS) [41], which includes ten affective symptoms commonly experienced during the PM period (e.g., “depression”). Again, each item was scored on a Likert scale ranging from 1 (not at all) to 5 (very much). A single score was computed by averaging the ten items. In our study, Cronbach’s alpha was 0.86. Higher scores indicated worse PM affective symptoms.

#### 2.4.3. PM Cognitive Resources

PM cognitive resources were assessed using the Self-Regulation Scale (SRS) [42], which comprises seven items aimed at evaluating perceived cognitive self-control (e.g., “I can concentrate on one activity for a long time, if necessary”) during the PM period. We preceded the items with the headline “During your premenstrual period…”. Each item was scored on a Likert scale ranging from 1 (not at all) to 5 (very much). A single score was computed by averaging the seven items (in our study, Cronbach’s alpha was 0.91), with higher scores indicating fewer PM cognitive symptoms (higher cognitive resources).

#### 2.4.4. Perceived Physical Ability

Perceived physical ability (PPA) was assessed via a validated Italian version of the Perceived Physical Ability Scale [43,44], which measures how confident an athlete feels about her/his physical ability, which is considered a good predictor of one’s actual sports performance. The scale consists of ten items (e.g., “Are you physically strong?”) measured on a Likert scale ranging from 1 (not at all) to 5 (completely). We preceded the items with the headline “During your premenstrual period…”. A single score of PPA was computed by averaging the ten items. In our study, Cronbach’s alpha was 0.89.

#### 2.4.5. World Health Organization Well-Being Index

Finally, well-being was assessed via the standardized Italian version of the five-items World Health Organization Well-Being Index (WHO-5) [45], a concise global rating scale measuring subjective well-being. The WHO-5 comprises five positive statements (e.g., “I have felt calm and relaxed”) that we preceded by the following headline: “Please indicate how often you experience the sensations described below during your premenstrual period”. Each of the five items was scored on a Likert scale ranging from 0 (never) to 5 (always). In our study, Cronbach’s alpha was 0.89.

### 2.5. Sample Size and Statistical Analysis

An a priori power analysis was run with G*Power [46]. This analysis found that a multiple regression analysis with eight predictors (five independent variables and three mediators), an alpha level of 0.05 (two-tailed), and a power of 0.80 to detect a small to medium effect size of ρ2 = 0.06, required a total sample size of N = 259. For this reason, a total of 263 participants were recruited to take part in this study.

The data were expressed as the means (M) ± standard deviation (SD). A series of one-sample *t*-tests was then used to define a significant deviation from the midpoint of the scale for each coping strategy. Furthermore, Pearson’s correlation was used to determine the relationships between the selected variables. A moderated mediation analysis was conducted using Hayes’ [47] PROCESS version 3.5 computational tool for SPSS (model 10). This tool enables the estimation of path coefficients, standard errors, and different indexes of effect size, as well as of the significance of the indirect effects obtained through the bootstrapping method with 5000 repetitions, with a confidence interval (CI) of 95% [48]. Statistical significance was set at *p* ≤ 0.05. Statistical analyses were processed using SPSS version 25.0 (IBM, Armonk, NY, USA).

## 3. Results

### 3.1. Participants

A total of 274 elite female athletes completed the questionnaire. Of these, 11 reported an irregular period (that is, a menstrual cycle shorter than 25 days or longer than 36 days) and were removed from the analyses. Therefore, N = 263 participants aged 16 years or more (M_age_ = 23.55; SD = 5.40) and active at the moment of the survey, that is, not retired or temporarily out of sports (e.g., due to a sports injury, motherhood), took part in the study.

There were 105 elite female rugby players with a good average rugby experience who filled the online questionnaire. Of these participants, ninety-eight did not have children, five had one child, while two had three children or more. Regarding the use of contraceptives, 77.1% did not use them, 14.3% took pills (oestrogen and progesterone), 2.9% took minipills (progesterone only), and the remaining 5.7% used a vaginal ring.

There were 158 Italian Serie A elite female football players with many years of football experience who took part in this study. Out of these participants, none had children, 89.9% did not use contraceptives, while the remaining 10.1% took pills or minipills (7%—oestrogen and progesterone; 3.1%—progesterone only).

The descriptive characteristics of the participants divided by sport are shown in Table 1.

### 3.2. T-Test and Correlations

The means, standard deviations, and Pearson’ correlation coefficients for the different measures are presented in Table 2. We used a one-sample *t*-test to determine if there was a significant deviation from the midpoint of the scale for each coping strategy. The use of the avoiding harm (*t*(263) = −7.22, *p* < 0.001), adjusting energy (*t*(263) = −16.01, *p* < 0.001), and communicating PM coping strategies (*t*(263) = −2.63, *p* < 0.01) were below the midpoint of the scale (3), while the use of the self-care PM coping strategy was in line with it (*t*(263) = −1.16, *p* = ns). Awareness and acceptance as a PM coping strategy were highly used (M = 4.15, SD = 0.60), as indicated by the fact that the average score was significantly different from the midpoint of the scale (*t*(263) = 31.02, *p* < 0.001). For all the variables, skewness and kurtosis were ≤ 1, indicating normal distribution of the data.

Looking at the differences between sports, the only significant ones were related to the PM symptoms experienced by the athletes (*p*s < 0.05) that were more invalidating for the rugby players (M = 3.08 for affective symptoms and 3.05 for physical symptoms) than for the football players (M = 2.88 for affective symptoms and 2.84 for physical symptoms).

Regarding the intervariable relations, avoiding harm as a PM coping strategy was positively correlated to both PM physical and affective symptoms and negatively correlated to PM cognitive resources. Moreover, avoiding harm as a PM coping strategy was negatively correlated to both PPA and well-being. The same pattern was shown for adjusting energy as a PM coping strategy. Awareness and acceptance as a PM coping strategies were positively correlated to PM cognitive resources and well-being, while self-care as a PM coping strategy, contrary to the hypothesis, was positively correlated to PM physical symptoms. Lastly, communicating as a PM coping strategy, in line with the hypotheses, was negatively correlated to PM affective symptoms and positively correlated to well-being.

### 3.3. Mediation Analysis

Based on the Pearson’s correlation results, two mediation model analyses were executed, considering the two dependent variables (i.e., PPA and well-being). In all the models, the five PM coping strategies (i.e., avoiding harm, awareness and acceptance, adjusting energy, self-care, and communicating) were the independent variables, while PM physical and affective symptoms and PM cognitive resources were the mediators. In Model 1, the dependent variable was the elite female athletes’ PPA, while in Model 2, the dependent variable was the elite female rugby and football players’ well-being. The results are shown in Table 3 (for PPA) and Table 4 (for well-being), as well as in Figure 1.

The avoiding harm PM coping strategy was indirectly associated with decreased PPA via decreased PM cognitive resources, but not via PM affective and physical symptoms; moreover, it was associated with decreased well-being via increased PM affective symptoms and decreased PM cognitive resources, but not via PM physical symptoms. A similar pattern was displayed by adjusting energy as a PM coping strategy, which was negatively associated with PPA directly (b = −0.17, SE = 0.06, 95% CI [−0.30, −0.05]), but also indirectly via increased PM affective and physical symptoms and via decreased PM cognitive resources. The use of adjusting energy was also negatively associated with well-being, both directly (b = −0.16, SE = 0.08, 95% CI [−0.317, −0.009]) and indirectly via increased PM affective symptoms and decreased PM cognitive resources, but not via PM physical symptoms.

The awareness and acceptance PM coping strategy was indirectly associated with increased PPA via increased PM cognitive resources and decreased PM physical symptoms, but not via PM affective symptoms; moreover, it was associated with increased well-being via increased PM cognitive resources, but not via PM affective and physical symptoms. The self-care PM coping strategy was indirectly associated with increased PPA via increased cognitive resources and decreased PM affective symptoms, but not via PM physical symptoms. Self-care was also indirectly associated with increased well-being, again, via increased PM cognitive resources and decreased PM affective symptoms, but not via PM physical symptoms. Lastly, communicating as a PM coping strategy was not associated with PPA, but was indirectly associated with increased well-being via decreased PM affective symptoms, but neither via PM physical symptoms nor PM cognitive resources.

In sum, the results of the mediation analyses supported the hypotheses. Specifically, they highlighted the dysfunctional impact of the avoiding harm and adjusting energy PM coping strategies on the elite female athletes’ PPA and well-being during their PM period. On the other hand, awareness and acceptance, self-care, and communicating were functional PM coping strategies. Furthermore, the mediators all intervened in the expected directions.

## 4. Discussion

We conducted a quantitative study aimed at explaining which coping strategies elite female rugby and football players use specifically during their premenstrual (PM) period to maintain perceived physical ability (PPA) and well-being. A mediation model analysis was run considering five PM coping strategies (i.e., avoiding harm, awareness and acceptance, adjusting energy, self-care, and communicating) as independent variables, PPA and well-being as dependent variables, and PM physical and affective symptoms and PM cognitive resources as mediators.

Coherently with H1 and H3, the avoiding harm PM coping strategy was indirectly associated with decreased PPA via lowered PM cognitive resources, but not via PM physical and affective symptoms. A similar pattern was displayed by the adjusting energy PM coping strategy; it was negatively associated with PPA directly, but also indirectly via increased PM affective and physical symptoms and decreased PM cognitive resources. The use of adjusting energy was also negatively associated with well-being both directly and indirectly via increased PM affective symptoms and decreased PM cognitive resources, but not via PM physical symptoms. These results could be explained by the specific condition of an elite female athlete who could perceive avoiding situations, activities, and people and adjusting energy, changing exercise habits as detrimental or dysfunctional for her PPA and well-being, also impacting or reducing her PM cognitive resources [40]. Indeed, while previous qualitative studies [24,49] showed these strategies were two of the more effective ones that the general female population can use to cope with PM changes and distress, for elite female athletes, it is not easy or possible to avoid potentially upsetting situations and people when more sensitive premenstrually. Indeed, if social withdrawal, avoidance of situations and persons, and refraining from exercise may not be maladaptive or dysfunctional for the general female population [25], they can be for elite female athletes who cannot always avoid situations and people causing PM distress and symptoms and reduce exercise because of the internal and external pressures to train, compete, and perform. These pressures may result in elite female athletes being more likely to endure training and competitions despite experiencing PM symptoms in comparison to the general female population or recreational and amateur female athletes [40].

Another possible explanation for elite female athletes not using avoiding harm as a way of coping is that they can cope with PM physical demands and pain because training and competing in high-level sports are intertwined with pain. Indeed, practicing sports at the top level may hurt, and elite female athletes could develop adaptive mechanisms.

Secondly, and in line with H2 and H3, our results showed that the awareness and acceptance PM coping strategy was indirectly associated with increased PPA via increased PM cognitive resources and decreased PM physical symptoms, but not via PM affective symptoms; moreover, the awareness and acceptance PM coping strategy was associated with increased well-being via increased PM cognitive resources, but not via PM affective and physical symptoms. A possible explanation of these results can be that being aware and accepting PM symptoms may enhance the attentional focus of elite female athletes on sporting tasks and empower their PM cognitive resources. Findlay et al. [25] found that while in the general female population PM symptoms generally increase distraction and reduce attentional focus and concentration in the work, life, and school domains, for elite female athletes, awareness and acceptance as a coping strategy is likely to prevent distraction from their work as athletes.

Moreover, elite female athletes seem to show greater PM cognitive resources and attentional focus during competitions and matches if compared with training [25]. It is unclear why this happens; however, the increased arousal and attentional focus during competitions and matches may override PM symptoms, a fact that does not occur during training. This, in turn, may increase the perception of well-being via increased PM cognitive resources, but not via PM affective and physical symptoms. Indeed, several psychological symptoms may affect elite female athletes in the PM period [40], as well as the general female population [50,51], manifesting higher worry, feeling of agitation, and negative mood states.

Accepting PM symptoms is not the only effective way of coping that elite female athletes may adopt as well as the general female population [52]. Indeed, self-care as a PM coping strategy in our study was indirectly associated with increased PPA via increased PM cognitive resources and decreased PM affective symptoms, but not via PM physical symptoms. Self-care as a PM coping strategy was also indirectly associated with increased well-being, again, via increased PM cognitive resources and decreased PM affective symptoms, but not via PM physical symptoms. A possible explanation for these results is that, in some cases, elite female athletes displayed an acceptance of their PM symptoms, reporting that they do feel that menstrual cycle issues are acceptable reasons to enhance self-care as a PM coping strategy even if they felt that they must continue to train and compete regardless of pain or other symptoms [25].

In addition, communicating as a PM coping strategy was not associated with PPA but was indirectly associated with increased well-being via decreased PM affective symptoms, but not via PM physical symptoms or PM cognitive resources. Compared with the general female population [52], Findlay et al. [25] found that elite female athletes are more accustomed to speaking to experts (e.g., medical, and technical staff members) and non-experts (e.g., teammates, friends, relatives) in the pursuit of coping with PM symptoms and maintaining well-being. However, some elite female athletes expressed concerns in approaching and communicating with their coaches about PM symptoms and distress. Most female athletes expressed a reluctance to confide in their coaches with these matters due to several reasons, including awkwardness and embarrassment, gender differences (because in the vast majority of the cases coaches are male), fear of being judged as fragile, and feeling like there would be nothing that the coach could do to help them [25]. It has been suggested that women are more likely to express their PM symptoms within comfortable environments (e.g., the home environment) [53]. Thus, elite female athletes could be more likely to disclose PM affective symptoms and distress when they perceive this not to negatively impact their PPA and well-being. In sum, H2 was confirmed, and awareness and acceptance, self-care, and communicating seem to be useful PM coping strategies because they indicate either positive disengagement or agency on the problem.

Third, we ran a mediation model analysis, considering PM coping strategies as independent variables, PPA and well-being as dependent variables, and PM physical and affective symptoms and PM cognitive resources as mediators. The results supported our third hypothesis (H3): a dysfunctional impact of avoiding harm (indirect) and adjusting energy (both direct and indirect) and a functional indirect influence of awareness and acceptance, self-care, and communicating as PM coping strategies on PPA and well-being were found during the PM period. As discussed before, avoiding harm as a PM coping strategy, that is, avoiding potential disturbing situations, people, or actions when more sensitive premenstrually, is not always possible for elite female athletes, as well as the use of adjusting energy as a PM coping strategy, for example, by reducing or adapting exercise. For these reasons, avoiding potentially harmful situations, people, or actions and adjusting energy are confirmed to be maladaptive or dysfunctional in terms of PPA and well-being for elite female athletes [25]. Conversely, awareness and acceptance, self-care, and communicating with experts and non-experts in a comfortable environment can serve as effective PM coping strategies with a functional impact on PPA and well-being [25].

### Limitations and Perspective

Notwithstanding the promising results, some limitations need to be acknowledged. First, we included in the study only elite female athletes practicing contact team sports (rugby and football). For this reason, further studies should include representative samples of elite female athletes practicing non-contact team sports (e.g., volleyball and baseball), but also individual contact sports (e.g., karate and judo) and individual non-contact sports (e.g., gymnastics and track and field) to evaluate potential differences on specific PM coping strategies depending on the type and characteristics of different sports (team vs. individual; contact vs. non-contact) that may have a great influence [54]. Second, we focused our attention specifically on the PM period, but future research should also consider the specific coping strategies that elite female athletes may use to cope with symptoms caused and related to other phases of the menstrual cycle, for example, the menstrual period itself. Another methodological limitation pertains to the selected measures. To the best of our knowledge, there is no standardized and validated questionnaire in Italian aimed at measuring PM symptoms and/or coping strategies (especially in the sports context). For this reason, we translated the Premenstrual Coping Measure (PMCM), the Daily Symptoms Rating (DSRS), and the Self-Regulation Scale (SRS) back-to-back for our research purposes. Future research should consider validating in Italian the aforementioned measures both in the general population and among female athletes. Lastly, we used a correlational design. Future studies might want to implement experimental studies (a) to establish the causal relationship between coping strategies and PM symptoms of elite female athletes and (b) to design and adopt training to promote the most functional coping strategies for this specific female population.

## 5. Conclusions

Women’s participation in sports has grown worldwide over the recent decades, including in sports that are typically associated with men (e.g., rugby and football). Correspondingly, research on female athletes has significantly increased, but some topics have been only scantly studied [7]: among these, the impact of the menstrual cycle and its effects on physical performance have been unaccounted for or very little studied [16,25] and remain fundamentally unknown [13,55], together with the most specific coping strategies that female athletes may use to cope with premenstrual (PM) physical and affective symptoms [31]. This study was aimed at quantitatively explaining which PM coping strategies elite female rugby and football players use during their PM period to maintain perceived physical ability (PPA) and well-being. As predicted, PM physical and affective symptoms as mediators reduced PPA and well-being, while PM cognitive resources enhanced them. These results may inform sports sciences practitioners (e.g., coaches, staff members, physical trainers, medical experts, sport psychologists, nutritionists, sleep consultants, etc.) on how to support elite female athletes in the pursuit of PPA and well-being.

## Figures and Tables

**Figure 1 ijerph-19-11168-f001:**
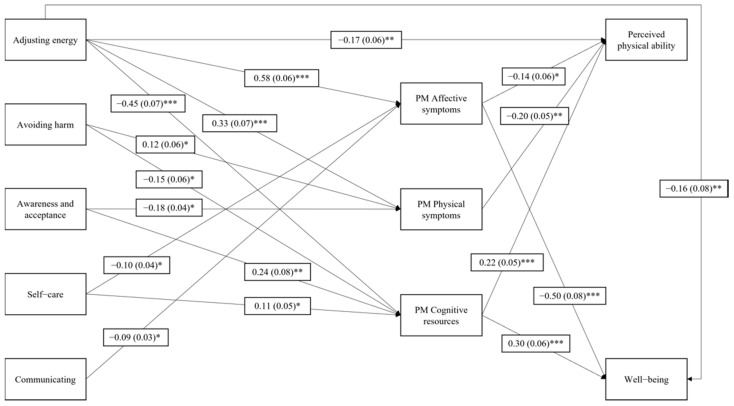
Mediation model of the effects of the five PM coping strategies via PM affective and physical symptoms and PM cognitive resources on perceived physical ability (PPA) and well-being. Only significant paths are reported (unstandarized coefficients; standard errors in parentheses; N = 263); * *p* < 0.05, ** *p* < 0.01, *** *p* < 0.001.

**Table 1 ijerph-19-11168-t001:** Descriptive characteristics of the sample.

Variables	Rugby (N = 105)		Football (N = 158)	
	M	SD	M	SD
Age (years)	25.78	6.17	21.33	4.63
Sports experience (years)	6.23	3.84	13.36	4.92
Deliberate practice (hours per week)	5.39	2.84	8.78	2.82
Contraceptives used (months)	50.38	39.87	23.75	21.28
Average length of the menstrual cycle (MC) (days)	29.23	10.64	28.86	6.79
Time from menarche (years)	13.43	6.33	8.45	4.54
Gender preferences regarding discussion of PM issues (−1 = male coach/staff member; +1 = female coach/staff member or teammates)	0.66	0.48	0.63	0.50

Note: M = mean, SD = standard deviation (N = 263).

**Table 2 ijerph-19-11168-t002:** Means (M), standard deviations (SD), and Pearson’ correlation coefficients for the study measures.

	M	SD	[β0]	[β1]	[β2]	[β3]	[β4]	[β5]	[β6]	[β7]	[β8]	[β9]
**[β0]** Avoiding harm	2.61	0.89	–	0.019	0.377 **	0.279 **	−0.145 *	0.357 **	0.269 **	−0.272 **	−0.278 **	−0.229 **
**[β1]** Awareness and acceptance	4.15	0.60		–	−0.082	0.220 **	0.292 **	−0.112	0.084	0.237 **	0.074	0.163 **
**[β2]** Adjusting energy	2.25	0.76			–	0.327 **	−0.006	0.569 **	0.347 **	−0.431 **	−0.464 **	−0.456 **
**[β3]** Self-care	2.93	1.03				–	0.205 **	0.076	0.160 **	−0.002	−0.110	0.022
**[β4]** Communicating	2.83	1.06					–	−0.177 **	−0.084	0.142 *	0.048	0.201**
**[β5]** PM affective symptoms	2.96	0.80						–	0.531 **	−0.511 **	−0.510 **	−0.646 **
**[β6]** PM physical symptoms	2.92	0.83							–	−0.281**	−0.446 **	−0.401 **
**[β7]** PM cognitive resources	3.36	0.85								–	0.479 **	0.545 **
**[β8]** Perceived physical ability (PPA)	3.57	0.74									–	0.515 **
**[β9]** Well-being	2.48	0.99										–

Note: For all the measures, the response scale ranged from 1 to 5, except for well-being (from 0 to 5); N = 263; * *p* < 0.05, ** *p* < 0.01.

**Table 3 ijerph-19-11168-t003:** Indirect effects in Model 1 with perceived physical ability (PPA) as the dependent variable.

Predictor	Mediator	Mean Bootstrap Estimate	Confidence Interval (95%)
Avoiding harm	PM affective symptoms	−0.02 (0.01)	[−0.052, 0.000]
Avoiding harm	PM physical symptoms	−0.03 (0.02)	[−0.055, 0.000]
Avoiding harm	PM cognitive resources	−0.03 (0.02)	[−0.067, −0.005]
Awareness and acceptance	PM affective symptoms	0.00 (0.01)	[0.024, −0.028]
Awareness and acceptance	PM physical symptoms	−0.04 (0.02)	[−0.024, −0.003]
Awareness and acceptance	PM cognitive resources	−0.05 (0.02)	[0.013, 0.108]
Adjusting energy	PM affective symptoms	−0.08 (0.04)	[−0.167, 0.001]
Adjusting energy	PM physical symptoms	−0.07 (0.03)	[−0.123, −0.025]
Adjusting energy	PM cognitive resources	−0.10 (0.03)	[−0.171, −0.042]
Self-care	PM affective symptoms	0.01 (0.01)	[−0.000, −0.037]
Self-care	PM physical symptoms	−0.00 (0.01)	[−0.026, 0.019]
Self-care	PM cognitive resources	0.02 (0.01)	[0.000, 0.151]

Note: Mean bootstrap estimates are based on 5000 bootstrap samples (standard error in parentheses) (N = 263).

**Table 4 ijerph-19-11168-t004:** Indirect effects in Model 2 with well-being as the dependent variable.

Predictor	Mediator	Mean Bootstrap Estimate	Confidence Interval (95%)
Avoiding harm	PM affective symptoms	0.08 (0.03)	[−0.145, −0.020]
Avoiding harm	PM physical symptoms	−0.01 (0.01)	[−0.042, 0.004]
Avoiding harm	PM cognitive resources	−0.04 (0.02)	[−0.067, −0.005]
Awareness and acceptance	PM affective symptoms	0.01 (0.04)	[−0.067, 0.085]
Awareness and acceptance	PM physical symptoms	−0.02 (0.02)	[−0.057, 0.002]
Awareness and acceptance	PM cognitive resources	0.07 (0.03)	[0.021, 0.137]
Adjusting energy	PM affective symptoms	−0.29 (0.06)	[−0.401, −0.182]
Adjusting energy	PM physical symptoms	−0.04 (0.02)	[−0.086, 0.009]
Adjusting energy	PM cognitive resources	−0.14 (0.04)	[−0.219, −0.068]
Self-care	PM affective symptoms	0.05 (0.02)	[0.004, 0.099]
Self-care	PM physical symptoms	−0.00 (0.01)	[−0.017, 0.012]
Self-care	PM cognitive resources	0.03 (0.02)	[0.001, 0.070]
Communicating	PM affective symptoms	0.05 (0.02)	[0.007, 0.092]
Communicating	PM physical symptoms	−0.01 (0.01)	[−0.004, 0.027]
Communicating	PM cognitive resources	−0.01 (0.02)	[−0.020, 0.046]

Note: Mean bootstrap estimates are based on 5000 bootstrap samples (standard error in parentheses) (N = 263).

## Data Availability

Not applicable.

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
