# Peer review of "How Do Elite Female Athletes Cope with Symptoms of Their Premenstrual Period? A Study on Rugby Union and Football Players’ Perceived Physical Ability and Well-Being"

_ijerph, 2022, doi:10.3390/ijerph191811168_

Round 1

Reviewer 1 Report

ijerph-1838882_review

Title: How do elite female athletes cope with symptoms of their pre-menstrual period? A study on Union Rugby and Football players’ perceived physical ability and well-being

Comments and Suggestions for Authors

Dear authors,

I was glad to have the opportunity to review this interesting manuscript that aimed to explain which coping strategies elite female rugby and football players use during their pre-menstrual period to maintain perceived physical ability and well-being. As has been well described in the article, the presence of women in sport in general worldwide is becoming increasingly important, also in those sports that have usually been associated with the male gender, such as rugby and football. Despite this, research related to the figure of women in sport continues to be a subject that is scarcely studied at present. Within this topic, the way in which the menstrual cycle impacts in the physical performance of female athletes and the coping strategies that these women can use to deal with their affective and premenstrual symptoms remain poorly studied. Therefore I agree with you that, despite the growing interest in this topic in recent years, many more studies are still needed to analyse how female athletes cope with premenstrual physical and affective symptoms. In this way, it would be possible to better understand how these aspects influence and interventions aimed at improving the performance and well-being of women who practice high-level sports could be established.

In my opinion, the introduction, objectives, methods, results, discussion and conclusions sections are well-described and analyses are appropriate. The bibliography is very complete and adequate.

I would like to make some minor comments that could be addressed to improve the document, in my opinion.

Specific comments:

Introduction

- In my opinion, the introduction is very complete; it contextualizes the topic in a very detailed way and highlights its importance. The key concepts of this manuscript, such as premenstrual coping strategies, perceived physical capacity and well-being are clearly and concisely presented. The objectives and hypotheses of the study are adequate.

Material and methods

- Page 5, lines 247-250. Could you add information about the time period in which the data was recorded? Where did the participants fill in the data? Did the participants complete the questionnaires at home or at their training centre? Was the registration done on a computer or through other devices such as tablets or mobile phones?

Results

- The results section is well-structured and comprehensive.

- Page 12, figure 1. Figure 1 is very interesting, congratulations.

Discussion

-Your discussion section is adequate and complete.

Conclusions

-Your conclusions are appropriate.

References list

- The bibliography is very complete.

I hope that my comments could help to improve the paper. Congratulations for your research.

Author Response

REVIEWER 1

Dear authors,

I was glad to have the opportunity to review this interesting manuscript that aimed to explain which coping strategies elite female rugby and football players use during their pre-menstrual period to maintain perceived physical ability and well-being. As has been well described in the article, the presence of women in sport in general worldwide is becoming increasingly important, also in those sports that have usually been associated with the male gender, such as rugby and football. Despite this, research related to the figure of women in sport continues to be a subject that is scarcely studied at present. Within this topic, the way in which the menstrual cycle impacts in the physical performance of female athletes and the coping strategies that these women can use to deal with their affective and premenstrual symptoms remain poorly studied. Therefore, I agree with you that, despite the growing interest in this topic in recent years, many more studies are still needed to analyse how female athletes cope with premenstrual physical and affective symptoms. In this way, it would be possible to better understand how these aspects influence and interventions aimed at improving the performance and well-being of women who practice high-level sports could be established.

In my opinion, the introduction, objectives, methods, results, discussion and conclusions sections are well-described and analyses are appropriate. The bibliography is very complete and adequate.

We would like to deeply thank the Reviewer for their endorsement and appreciation for our work.

I would like to make some minor comments that could be addressed to improve the document, in my opinion. Specific comments:

Introduction

- In my opinion, the introduction is very complete; it contextualizes the topic in a very detailed way and highlights its importance. The key concepts of this manuscript, such as premenstrual coping strategies, perceived physical capacity and well-being are clearly and concisely presented. The objectives and hypotheses of the study are adequate.

We thank the Reviewer for their positive feedback.

Material and methods

- Page 5, lines 247-250. Could you add information about the time period in which the data was recorded? Where did the participants fill in the data? Did the participants complete the questionnaires at home or at their training centre? Was the registration done on a computer or through other devices such as tablets or mobile phones?

We thank the Reviewer for the suggestion. We added the requested information in the Materials and Methods section of the paper (see p. 4, lines 182-186).

Results

- The results section is well-structured and comprehensive.

- Page 12, figure 1. Figure 1 is very interesting, congratulations.

We thank the Reviewer for the appreciation.

Discussion

-Your discussion section is adequate and complete.

Conclusions

-Your conclusions are appropriate.

References list

- The bibliography is very complete.

Again, we thank the Reviewer for the positive feedbacks.

I hope that my comments could help to improve the paper. Congratulations for your research.

We thank the Reviewer for the appreciation and for the useful suggestions.

Reviewer 2 Report

Dear authors

I congratulate the authors of this work, especially because the topic is very useful for coaches, strength and conditioning and researchers in general.  However, at the current moment your works should be improved before being considered for publication.

Please check my comments:

Abstract

Please remove the headings to follow journal's guidelines

L32-33 - Please explain how

Keywords - Please avoid repeating the same words that are present in the title.

Introduction – introduction is too long, and it should be reduced in about one page. Two pages will be enough. Please check the repeated information or information that does support the aim of your topic. The following comments will provide some suggestions.

L58-72 - information too general. It seems to repeat some information of the previous paragraphs. It suggested to remove it

L73-81 - Again, why are the authors stating differences between sex if the study was only conducted with females? For instance, the next paragraph is much more important, and it is the main topic of your work.

L91-97 - references are needed here.

L98-100 - again, references are needed here. At least two.

L154-157 - references are needed here.

L158-160 - it is suggested to provide the aims of the study in the end of the introduction. Some hypotheses should be added.

L164-182 - These concepts can be explained in introduction. However, references are needed. Considering the way, they are described, another suggestion could be to move this information to the methods section.

2. Aim of the study - this section should be merged with introduction without repeating information.

Materials and Methods

It is suggested to present study design and instead of eligibility criteria, use Participants. In addition, please add participants information as well.

L262 - It is suggested to explain that questionnaire was applied in Italian.

L272 - was this translated to Italian or applied in English? Please explain. If it was translated, did you validate it?

Please explain these questions to the remaining questionnaires with the exception of Physical Activity Scale that according to your reference, it was applied in Italian.

L332 – “range” - minimum and maximum? Please clarify.

L332-333- did you check normality of data? Differences between groups? This was not explained in the methods section, neither in the results section. Please clarify

Results

L353 - this data does not match data from table 1. Moreover, it is suggested to remove it from the text to avoid repetition

Table 1 - range data was not provided here as mentioned in the statistical analysis section

Section 4.2 - where are results from T-test?

Table 2 - instead of [1] [2] and so on, it is suggested to change for β0 β1 β2 etc. Also, please reduce space between lines.

Discussion

The 2 first paragraphs should explain the aims and the main results. The authors only present their results on the 3rd. Thus, it is suggested to change this section for better clarity. Then maybe helpful to organize your discussion in 3 subsections according to your hypothesis

Thank you

Author Response

REVIEWER 2

Dear authors,

I congratulate the authors of this work, especially because the topic is very useful for coaches, strength and conditioning and researchers in general.  However, at the current moment your works should be improved before being considered for publication.

We would like to thank the Reviewer for their endorsement and insightful feedback on the paper. In the next lines, a point-by-point answer to the issues and requests made is presented. We hope the Reviewer will find the revision satisfying.

Please check my comments:

Abstract

Please remove the headings to follow journal's guidelines

We removed the headings from the Abstract (see p. 1, lines 19-33)

L32-33 - Please explain how

We added a brief explanation at current p. 1, line 33 (…by knowing and reinforcing the most functional PM coping strategies for them).

Keywords - Please avoid repeating the same words that are present in the title.

Thank you for the suggestion. We modified the keywords in: Coping strategies; pre-menstrual symptoms; athletes’ wellness; team sports (p. 1, line 34).

Introduction – introduction is too long, and it should be reduced in about one page. Two pages will be enough. Please check the repeated information or information that does support the aim of your topic. The following comments will provide some suggestions.

L58-72 - information too general. It seems to repeat some information of the previous paragraphs. It suggested to remove it

L73-81 - Again, why are the authors stating differences between sex if the study was only conducted with females? For instance, the next paragraph is much more important, and it is the main topic of your work.

Thank you for the suggestion. We shortened the Introduction (see p. 1-3, lines 38-154) to a bit more than two pages including the now merged “Aim of the study” section. We also reduced the information about differences between sexes. We left, however, the argument regarding gender mainstreaming because it is the perspective this study applies, which is oriented to considering specific needs of female athletes that have for a long time been neglected both by researchers and by practitioners by moving from the (incorrect) idea that they were like those of male athletes.

L91-97 - references are needed here.

L98-100 - again, references are needed here. At least two.

L154-157 - references are needed here.

We added appropriate references to all the presented information (see p.2, line 71, new References 15 and 16; see p. 3 line 115 new Reference 28).

L158-160 - it is suggested to provide the aims of the study in the end of the introduction. Some hypotheses should be added.

We moved the indicated lines in the (former) Aim of the study section, where also the hypotheses are presented (see new p. 3, lines 117-119 and p. 3, lines 138-154).

L164-182 - These concepts can be explained in introduction. However, references are needed. Considering the way they are described, another suggestion could be to move this information to the methods section.

We agree with the Reviewer: we shortened the description of each coping strategy and moved them to the Materials and Methods section (see p. 4-5, lines 195-207).

  1. Aim of the study - this section should be merged with introduction without repeating information.

Thank you for the suggestion. We merged the two sections (former Aim of the study and Introduction) and removed the redundant information.

 Materials and Methods

It is suggested to present study design and instead of eligibility criteria, use Participants. In addition, please add participants information as well.

Accordingly with the Journal’s criteria and the CROSS guidelines (https://www.researchgate.net/publication/353522424_Checklist_for_Reporting_Of_Survey_Studies_CROSS), we presented the Eligibility criteria in the Materials and Methods section, and the Participants information in the Results section.

L262 - It is suggested to explain that questionnaire was applied in Italian.

Thank you, we added the disclaimer, please see p. 4, line 182 (The online questionnaire was administered in Italian to…).

L272 - was this translated to Italian or applied in English? Please explain. If it was translated, did you validate it? Please explain these questions to the remaining questionnaires with the exception of Physical Activity Scale that according to your reference, it was applied in Italian.

Thank you for the opportunity to clarify this. As now specified on p. 4, line 182, we administered the questionnaire in Italian. The Perceived Physical Ability Scale and the WHO-5 Index were both used in their Italian validated form. Given that there is no standardized and validated questionnaire in Italian aimed at properly measuring PM coping strategies, as well as PM symptoms, we back-to-back translated the Pre-Menstrual Coping Measure (PMCM), the Daily Symptoms Rating (DSRS), and the Self-Regulation Scale (SRS) scales and calculated their internal consistency within our sample, which proved to be very high for all measures. Of course, these measures are not validated in Italian, and we now acknowledged this limitation on paragraph 5.1. Limitations and Perspective (please see p. 12, lines 484-489).

L332 – “range” - minimum and maximum? Please clarify.

We removed the reference to “the range”, given that we only provided means and SDs in the Tables. Thanks for suggesting.

L332-333- did you check normality of data? Differences between groups? This was not explained in the methods section, neither in the results section. Please clarify

We added a line presenting information to the skewness and kurtosis of data (please, see p. 7, lines 299-300). Thank you for the suggestion.

For all variables, skewness and kurtosis were ≤ 1, thus suggesting a normal distribution of data.

We also added information about the differences between sports (please, see p. 7, lines 301-304).

Looking at the differences between sports, the only significant ones were related to the PM symptoms experienced by the athletes (both ps < .05), that were more invalidating for rugby players (M = 3.08 for affective symptoms, M = 3.05 for physical symptoms) than for football players (M = 2.88 for affective symptoms, M = 2.84 for physical symptoms).

Results

L353 - this data does not match data from table 1. Moreover, it is suggested to remove it from the text to avoid repetition

Thank you very much for noticing! The mistake was in the text but, in accordance with this indication, we removed it to avoid repetitions.

Table 1 - range data was not provided here as mentioned in the statistical analysis section

Thank you for noticing. As stated above, we do not believe that it is a fundamental information (especially considering the information and analyses added to Paragraph 4.2), therefore we removed the indication in the text.

Section 4.2 - where are results from T-test?

We used a one-sample t-test to determine if there was a significant deviation from the mid-point of the scale for each coping strategy. We added this explanation on p. 7, lines 297-298.

We used a one-sample T-test to determine if there was a significant deviation from the mid-point of the scale for each coping strategy.

Table 2 - instead of [1] [2] and so on, it is suggested to change for β0 β1 β2 etc. Also, please reduce space between lines.

We made all the requested changes, thank you for suggesting them.

Discussion

The 2 first paragraphs should explain the aims and the main results. The authors only present their results on the 3rd. Thus, it is suggested to change this section for better clarity. Then maybe helpful to organize your discussion in 3 subsections according to your hypothesis

We deeply thank the Reviewer for this feedback. As suggested, we modified the Discussion section (please, see p. 10 from line 369). Now, we hope that the Discussion is more fluent and easier to follow now (see p. 10-12, lines 369-468).

Reviewer 3 Report

Thank you for submitting your valuable manuscript to this journal.

The aim of the present study was to find coping strategies of female athletes during PM period to maintain PPA and well-being.

There are some major points that may improve the study report for publication:

1-      The introduction of the manuscript is too long with lots of redundant data about the increasing number of women who participate in rugby and football. The international Olympic committee decision about gender equity is unrelated to the aims of this study. Moreover, too many previous studies are mentioned on introduction. The optimum length of the introduction part including the aim of the study is one to one and half page.

2-      The introduction of the manuscript starts whit “even if” which is not appropriate for a scientific paper.

3-      One important inclusion criteria was being elite female athlete. As you know, menstrual cycle irregularity is common among this population. How did you deal with this problem? There is no data in the results in this regard!

4-      Were the questionnaires used to measure study variables valid and reliable in Italian language? It must be mentioned and referenced in the manuscript. Cronbach’s alpha in your study is not sufficient for this purpose.

5-      Scoring the questionnaires and reporting the results are not clearly explained. Averaging the scores of different items in a questionnaire is not appropriate and you cannot compute a single score by averaging the scores from different questionnaires with various scales!

6-      Line 322 mentions that the WHO well-being index range is within 0 to 25, but on table 2 it is ranging from 0 to 5! You cannot change the scale of a questionnaire scores.

7-      I’m not sure that the T-test is appropriate here! It is used to determine if there is a significant difference between the means of two groups.

Author Response

REVIEWER 3

Thank you for submitting your valuable manuscript to this journal.

The aim of the present study was to find coping strategies of female athletes during PM period to maintain PPA and well-being. There are some major points that may improve the study report for publication:

We would like to sincerely thank the Reviewer for their detailed feedback on the paper. In the next lines, a point-by-point answer to the issues made is presented.

1-      The introduction of the manuscript is too long with lots of redundant data about the increasing number of women who participate in rugby and football. The international Olympic committee decision about gender equity is unrelated to the aims of this study. Moreover, too many previous studies are mentioned on introduction. The optimum length of the introduction part including the aim of the study is one to one and half page.

We warmly thank the Reviewer for this suggestion. Based on it, and based also on Reviewer 2’s opinion, we significantly shortened the Introduction (see p. 1-3, lines 38-154). It is now slightly more than two pages, but - based on Reviewer 2’s request - it also includes the former “Aim of the study” section. We removed the paragraph on the international Olympic committee decision about gender equality and reduced the number of studies mentioned.

2-      The introduction of the manuscript starts whit “even if” which is not appropriate for a scientific paper.

As suggested, we changed the opening, as can be seen on p. 1, beginning of line 38.

3-      One important inclusion criteria was being elite female athlete. As you know, menstrual cycle irregularity is common among this population. How did you deal with this problem? There is no data in the results in this regard!

We thank the Reviewer for the opportunity to clarify this. A menstrual cycle is considered “regular” when ranging from 25 to 36 days. All the analyses were performed on those athletes who reported a regular menstrual cycle. The period length in our final sample was M = 29,55 days (SD = 2,80). We added this information on p. 6, lines 270-272.

A total of 274 elite female athletes completed the questionnaire. Of these, 11 reported an irregular period (that is, a menstrual cycle shorter than 25 days or longer than 36 days) and were removed from the analyses.

4-      Were the questionnaires used to measure study variables valid and reliable in Italian language? It must be mentioned and referenced in the manuscript. Cronbach’s alpha in your study is not sufficient for this purpose.

Thank you very much for the question. As we describe in the Materials and Methods section, both the Perceived Physical Ability Scale and the WHO-5 Index were used in their Italian validated form. Unfortunately, there is no standardized and validated questionnaire in Italian aimed at properly and specifically measuring PM coping strategies, as well as PM symptoms (especially in the sport context). Therefore, we back-to-back translated the Pre-Menstrual Coping Measure (PMCM), the Daily Symptoms Rating (DSRS), and the Self-Regulation Scale (SRS) scales and calculated their internal consistency within our sample, which proved to be very high for all measures. We acknowledged this as a limitation of our study (please, see p. 12, lines 484-489).

To the best of our knowledge, there is no standardized and validated questionnaire in Italian aimed at measuring PM symptoms and/or coping strategies (especially in the sport context). For this reason, we back-to-back translated the Pre-Menstrual Coping Measure (PMCM), the Daily Symptoms Rating (DSRS), and the Self-Regulation Scale (SRS) scales for our research purposes. Future research should consider to validate in Italian the aforementioned measures both in the general population and among female athletes.

5-      Scoring the questionnaires and reporting the results are not clearly explained. Averaging the scores of different items in a questionnaire is not appropriate and you cannot compute a single score by averaging the scores from different questionnaires with various scales!

Thank you for the opportunity to clarify this point. All questionnaires were unidimensional; therefore, averaging the items was the appropriate way to obtain the final measure and in line with previous research. The only exceptions were:

  • the Pre-Menstrual Coping Measure (PMCM), which has a different subscale for each coping strategy, which we computed accordingly;
  • the Daily Symptoms Rating (DSRS) from which, accordingly with the different subscales, we derived two different measures, that is the PM Physical symptoms and the PM Affective symptoms.

6-      Line 322 mentions that the WHO well-being index range is within 0 to 25, but on table 2 it is ranging from 0 to 5! You cannot change the scale of a questionnaire scores.

Thank you for noticing this. In the clinical context, the WHO Well-Being Index is scored by summing the items to individuate thresholds useful to diagnosis. This is not our case (and of many other studies): we are using the scale for research purposes within a correlational study. Given that, the mean of individual item scores is perfectly correlated with the sum of the item scores, for correlations and regressions it makes no difference whether you use the mean or sum of items. We preferred to use the mean of items scores because the interpretation of a mean is clearer than a sum for the reader, and to be coherent with the other variables scores, which were expressed as means. We however agree with the Reviewer that (former) line 322 was confusing with respect to Table 2, so we removed the reference to the sum-score in the Materials and Methods section.

7-      I’m not sure that the T-test is appropriate here! It is used to determine if there is a significant difference between the means of two groups.

In this case, we did not use the paired sample t-test to determine if there is a significant difference between the means of two groups. We used a one-sample t-test to determine if there was a significant deviation from the mid-point of the scale for each coping strategy. We added this explanation on p. 6, lines 292-293.

Round 2

Reviewer 2 Report

Dear authors,

Congratulations for the revised work. I only have one comment. 

Previously I suggested to present study design first, and instead of eligibility criteria, to use Participants. The authors answered that “

“Accordingly with the Journal’s criteria and the CROSS guidelines (https://www.researchgate.net/publication/353522424_Checklist_for_Reporting_Of_Survey_Studies_CROSS), we presented the Eligibility criteria in the Materials and Methods section, and the Participants information in the Results section.”

However, you check the CROSS guidelines, the order of the section is still not fine. Study Design is still needed as well as participants section (Sample characteristics).

Appart from this comment, I believe that your manuscript has conditions to be accepted. 

Thank you

Author Response

REVIEWER 2

Dear authors,

Congratulations for the revised work. I only have one comment. 

We would like to thank the Reviewer for the comment.

Previously I suggested to present study design first, and instead of eligibility criteria, to use Participants. The authors answered that “

“Accordingly with the Journal’s criteria and the CROSS guidelines (https://www.researchgate.net/publication/353522424_Checklist_for_Reporting_Of_Survey_Studies_CROSS), we presented the Eligibility criteria in the Materials and Methods section, and the Participants information in the Results section.”

However, you check the CROSS guidelines, the order of the section is still not fine. Study Design is still needed as well as participants section (Sample characteristics).

Appart from this comment, I believe that your manuscript has conditions to be accepted. 

Thank you

As suggested, we added to the Materials and Methods section a new paragraph 3.1 in which we present the Study design (see p. 4, lines 158-161), and again as suggested we added at paragraph 3.3 the Sample characteristics. We would like to thank the Reviewer for the comment that has improved our manuscript.

Reviewer 3 Report

Thank you for the revision of your valuable manuscript.

The manuscript format is now acceptable for publication, but the introduction of the paper is still too long. I suggest to remove hypotheses H1-3 and explain the main purpose of the study in just one sentence.

Good luck.

Author Response

REVIEWER 3

Thank you for the revision of your valuable manuscript.

The manuscript format is now acceptable for publication, but the introduction of the paper is still too long. I suggest to remove hypotheses H1-3 and explain the main purpose of the study in just one sentence.

Good luck.

We would like to sincerely thank the Reviewer for the feedback on the paper. We agree that the Introduction is not short but at the same time we believe that it is complete. We thank the Reviewer for the suggestion to remove H1-3 and add the purpose of the study in one sentence, but at the same time we think that having clear and separated Hps is an added value for the manuscript. In addition, we also used the Hps for the Discussion that now we think complete and clear. For all these reasons, we prefer to have the H1-3 in the text. We would like to thank the Reviewer for the comment.